# Strategies to improve patient loyalty and medication adherence in Syrian healthcare setting: The mediating role of patient satisfaction

**Firas AlOmari**[ID]*[◉], **Abu Bakar A. Hamid**[◉]

Department of Marketing, Putra Business School, Universiti Putra Malaysia, Serdang, Selangor, Malaysia

◉ These authors contributed equally to this work.
* fomari6@gmail.com

**Data Availability Statement:** The relevant data are within the paper and its Supporting Information files.

## Abstract

The purpose of this study is to empirically examine the relationships between service quality, patient satisfaction, patient loyalty and medication adherence in the Syrian healthcare setting from a patient's perspective. Based on random sampling technique, data collection was conducted in six hospitals located in the Syrian capital Damascus. The reliability and validity of the theoretical model had been confirmed using quantitative analyses SmartPLS software. The study indicated that our proposed model can significantly explain (35) per cent of patient satisfaction, (55) per cent of patient loyalty and (46) per cent medication adherence in a statistically manner. Our results highlighted that patient satisfaction mediated the relationship between patient loyalty and service quality (assurance, reliability and financial aspect). Besides, patient satisfaction had mediation effect on the relationship between medication adherence and service quality (reliability and financial aspect). Financial aspect had the highest impact on patient satisfaction (β = 0.242) and medication adherence (β = 0.302). In addition, reliability was the only dimension of service quality that had a significant direct impact on patient satisfaction, patient loyalty and medication adherence. To increase patient loyalty in Syrian hospitals, healthcare professionals should place a greater emphasis on the reliability and responsiveness elements of service quality. To the author's knowledge, this is the first study conducted during the COVID pandemic to evaluate the mediating role of patient satisfaction in the relationship between service quality, patient loyalty and medication adherence in the Syrian healthcare sector.

## 1. Introduction

The emergence of the COVID pandemic led to an unprecedented need to change human behaviour across the whole population in a very short period of time [1]. The World Health Organization (WHO) has been warned that during the COVID pandemic, anxiety, and stress have increased worldwide remarkably [2]. The global epidemic crisis has been a critical period

**Funding:** The authors received no specific funding for this work.

**Competing interests:** The authors have declared that no competing interests exist.

for international and national economies as well as household finances. The devastating economic impact of COVID has a negative impact on the world economy and on all industries. In 2011, the unrest in Syria led to an unprecedented health catastrophe and an increment in communicable and non-communicable diseases [3] that extended across several countries in the region, and this condition has adversely affected the economy and healthcare systems [4]. It is estimated that a budget of USD 1.5 billion will be required to rebuild Syria's healthcare system [5]. Syrian Ministry of Health announced that there is a need to assess healthcare infrastructure, healthcare technology, maintenance capacities, medical consumables as well as the quality-of-care services in public and private healthcare hospitals [5, 6]. There is widespread evidence that the failure to comply with medication therapy is a longstanding problem that is connected to inadequate communication between the patient and the service provider, leading to disease development, reduced quality of life [7], and increasing the costs of healthcare [8, 9]. In addition, poor compliance behaviour is the most common cause of non-responding behaviour to the medication process. It has been proven that patients who comply with medical treatment plans have better health outcomes than those who do not, even when taking a placebo [10]. A customer's non-compliance behaviour has a negative impact on the quality of the services delivered by the service provider [11].

This manuscript organized in eight sections. In the first section, a literature review about the relationships between service quality, patient satisfaction, patient loyalty and medication compliance will be presented. Besides, underpinning theories that support our conceptual model will be highlighted. In this study, we implemented expectation confirmation theory (ECT) and health belief model (HBM) to explain the relationships between independent variables, the mediator, and the dependent variables. The second section will illustrate the proposed hypotheses and the conceptual framework. The third section presents research methodology that includes the sample size and the proposed measurement instruments. The fourth section will address the assessment of measurement model and structural model. The fifth section will highlight the findings and discussions. This section will present and discuss the results of the proposed hypotheses. The limitations and future work will be addressed in the sixth section. The next section will present the implications and recommendations for improving healthcare outcomes in Syria. Section eight reflects the main conclusion of this study.

## 2. Literature review

This section will present the previous research studies related to service quality, patient satisfaction, patient loyalty and medication compliance.

### 2.1 Relationship between service quality patient satisfaction and patient loyalty

The concept of service quality is a multidimensional, higher order construct [12, 13]. Donabedian [14] defined healthcare quality as "the application of medical science and technology in a manner that maximises its benefit to health without correspondingly increasing the risk". Parasuraman et al. [12] constructed a SERVQUAL model to evaluate service quality by comparing expectations with perceptions of ten dimensions (access, communication, competence, courtesy, credibility, tangibles, reliability, responsiveness, security and understanding customers). The SERVQUAL still a generic tool with good reliability and validity [15], and has been widely applied in different sectors including healthcare industry [16, 17]. Till now there is no consensus about how many dimensions should be used to measure service quality in healthcare organization. For example, some researchers used three dimensions [18], four dimensions

[19], five dimensions [20], six dimensions [21], seven dimensions [22], eight dimensions [23], ten dimensions [24]. Since the findings were unclear and still some ambiguity on the scale to measure service quality, there is a need to further investigate the nature of service quality dimensions in healthcare setting.

Based on an extensive literature review, we proposed a service quality construct that was originally adapted from the SERVQUAL model. We slightly modified with minor alteration in the wording of items to adapt them to healthcare setting and to fit local perceptions (Syrian healthcare setting). Currently, most Syrians face financial pressure, so it is difficult to adjust their budget and lifestyle by increasing expenses and decreasing income [25–27]. Therefore, there was an essential need for further research to uncover the influence of financial aspect on healthcare outcomes. The financial aspect dimension was added to the service quality construct, in order to gain a broader understanding of healthcare service quality in the Syrian hospitals. Javed et al. [28] indicated that a modified SERVQUAL scale must include more vital factors, especially when research is conducted in low-medium countries, such as service price/cost that is related to the service provider's care cost efficiency. Besides, Jennings et al. [29] highlighted that there is a gap in assessing the effectiveness of healthcare services provided by medical professional on waiting times, cost and patient satisfaction. Alumran et al. [30] indicated that patient's financial concerns need to be investigated that has impact on the improvement of the overall quality of health care services at Saudi Arabia's health care system. There is a need to included the financial aspect component to address the gap in assessing patients' perceptions of care costs [28–30]. In this study, a financial aspect variable that was not included in the original SERVQUAL model will be added to acquire a better knowledge of healthcare service quality in Syrian hospitals. Based on the development of questionnaire, the service quality construct has six dimensions (tangibility, empathy, assurance, reliability, responsiveness and financial aspect).

Healthcare providers should understand patient needs in order to achieve the highest level of satisfaction as it is the main goal of quality health services [31]. Patient satisfaction is an important concept in determining the healthcare services and considered as a major achievement indicator in healthcare organization [32]. In general, customer will be satisfied if perceived performance meets or exceeds customer's expectations, otherwise will dissatisfied [15, 33]. Patient satisfaction, according to Crowe et al. [34], is a cognitive process. Nevertheless, there is no consensus on a conceptual model that provides a holistic view of patient satisfaction and explains how customers become satisfied or unsatisfied. Tse and Wilton [35] gave a definition to customer satisfaction as the customer's reaction to the evaluation of the perceived discrepancy between customer's expectation and the real performance of the provided service after purchasing. Chahal and Mehta [36] in their study considered patient satisfaction a multidimensional construct consists of four dimensions, while other researchers designed and tested their own instrument to identify factors that predict satisfaction in the doctor patient relationship [37]. Customer loyalty leads to increase profitability that positively influencing firm financial performance, product-marketplace performance and generates shareholder wealth [38, 39]. Loyal customers are customers who hold favourable behaviours toward the service provider, committing to repurchase the service or product, and recommend it to others [40]. Customers may be loyal due to lack of alternatives service provider in the market or high switching obstacles. However, customers may also be loyal when they are satisfied and therefore want to continue the relationship [41]. Furthermore, doctors' interaction behaviour plays a remarkable role in building an effective relationship with their patients and increasing patients' confidence in medical professionals, as well as improving patients' loyalty to care service providers [42].

Many studies have attempted to establish the linkage between service quality (SQ) and patient satisfaction (PS), in addition to investigate their effects on patient loyalty (PL). Previous

studies highlighted that service quality has a significant effect on patient satisfaction [43, 44]. Amin and Nasharuddin indicated that there is no consensus concerning the relationship between patient satisfaction and service quality in the hospital industry [45]. Service quality dimensions have not the same impact on patient satisfaction and patient's behavioural intention [46–51]. Perceived service quality mediates the relationship between customer satisfaction and customer loyalty intention [52, 53]. Kondasani and Panda [54] found that the quality of amenities (tangible dimension) positively impacts patients' loyalty and satisfaction. Service quality dimensions (responsiveness, empathy and assurance) impact patient loyalty, while tangibility and reliability have not any influence on loyalty [55]. The doctor's expertise encourages patients to repurchase the same service, which increases customer retention [56]. Eleuch [57] indicated that patient loyalty is influenced by technical quality standards, customer's first impression of the hospital staff and care services. According to another study, however, found that while service quality had no direct effect on loyalty intentions, it had a significant indirect effect on loyalty intentions through patient satisfaction [58]. Based on this discussion, the findings are still mixed and contradictory, and therefore there is a need to investigate further the relationship between service quality, patient satisfaction and loyalty. One of the main contributions of our study is providing the empirical evidence of the mediation role of patient satisfaction on the relationship between service quality dimensions (tangibility, empathy, assurance, reliability, responsiveness and financial aspect) and patient loyalty from patient's perspective in healthcare sector in Damascus.

## 2.2 Relationship between service quality patient satisfaction and medication compliance

Compliance has been investigated from a wide range of scientific perspectives such as psychology, medical sciences and economics. At present, there is no agreement about a commonly accepted definition of compliance that causes to use other terms, such as, therapeutic alliance, concordance, co-operation, self-management and adherence. In addition, there is a deficiency of reliability and consistency in the measuring of compliance in the healthcare setting [59]. Medication compliance has become a very important topic since the clinical regimens are useless if patients do not comply with doctor advice [60]. Patient role in treatment procedure had transformed from passive recipient to a respected player that share and participate in decision making process with healthcare provider [61, 62]. Noncompliance can take different forms, for example, failing to fill prescription, taking less or more medicine than prescribed by medial professional, taking other patient medication, failing to comply with time. The quality of the patient-doctor relationship considered as an essential step of intervention to develop patients' adherence [63]. A lack of effective communication between healthcare professionals and patients leads to poor disease control, depression, sadness, noncompliance behaviour and patient dissatisfaction [64]. There is a significant relationship between compliance and service quality [65]. Rhodes [66] indicated that a good interaction and communication behaviour between physicians and patients in emergency room leads to increase patient adherence behaviour. In addition, deficiency to guarantee transferring adequate information in a proper way to patients regarding their medical status could result in deterioration of mental and physical health that progressively worse over time.

Effective communications skills and trust leads to better care management, improved perceived quality of healthcare services, higher level of patient satisfaction, better compliance to the treatment [67]. Failing to clarify the benefits and side effects of taking drugs, prescribing complex drug regimens, and unserious considering the financial obstacles that patients may suffer considered reasons to non-compliance [68, 69]. Understanding the forces driving

patient satisfaction will improve the patient's clinical course and adherence to medical treatment [70]. When patients become dissatisfied with the quality of interaction process with medical staff such as doctors, they are less likely to cooperate with them or even listen to them [71, 72]. The lack of consultation time may impede engaging the patient in a discussion on the importance of adherence to medication regimens. Providing a rationale may convince patients to inform their doctors about their non-adherence behaviour and allowing for discussion [73].

Some consumers will pick a higher-priced brand over a lower-priced one when there is substantial doubt about the service quality level between two service providers. This implies that these customers judge quality based on pricing [74]. There is strong evidence that the economic crisis has led to reductions in the use of routine medical care [75, 76]. During financial shortfalls at hospitals and when quality is imperfectly observed by patient, financially constrained healthcare providers have incentives to lower their product quality to increase cash flows in the short run [77]. Patients who take multiple medications may suffer from a financial burden, especially in these days where there is a drop in spending due to a recession and fluctuation in Syrian economy. Furthermore, the proportion of patient compliance is often low if people do not redeem their prescriptions or do not take drugs as directed by medical practitioners [78]. Kennedy et al. [8] highlighted that non adherence to the medication increases the costs of health care. Whereas, other studies [79, 80] showed that healthcare cost of adherent patients had been increased in comparison to non-adherent patients. Poor compliance to medication regimens contributing to considerable deteriorating of disease [68]. There is still a deficit of reliability and consistent measurement of healthcare compliance [59] and lack of an underlying theoretical framework in various studies on selecting compliance-related independent factors [81]. Moreover, understanding the main obstacles to adherence behaviour may help healthcare providers in Syrian hospitals to determine probable interventions to increase patient behavioural compliance to improve patients' lives. This study has gone some way towards improving understanding of the mediating effect of patient satisfaction on the relationship between service quality, and medication adherence in Syrian healthcare setting.

## 2.3 Underpinning theories

In this research work, were employed expectation confirmation theory (ECT) and health belief model (HBM) as underpinning theories to clarify the relationships between service quality, patient satisfaction, patient loyalty, and medication adherence.

**2.3.1 Expectation Confirmation Theory (ECT).** Expectation Confirmation Theory (ECT) provided by Oliver [82]. Based on this theory, customer response to the service quality was caused by comparing between actual services with what customer expected from the service provider. Therefore, patient satisfaction happened when patient's perceived service quality is more than their expectation. In the marketing literature, Expectation–Confirmation Theory (ECT), also known as Expectation–Disconfirmation Theory (EDT), has been used to demonstrate the repurchase intention of the consumers for many different products and services including durable and nondurable products [83]. Expectation refers to an individual's prediction of consumption of a particular service/product that might happen in the future [84]. Based on expectation-confirmation theory, Huang indicated that satisfaction wielded the strongest endogenous influence on continued use [85]. The expectation-confirmation theory (ECT) stands out as being robust for modelling repurchase behaviour and recommendation intention in marketing research [86] of which are important components of loyalty [87].

**2.3.2 Health Belief Model (HBM).** There is a deficiency of reliability and consistency in the measuring of compliance in healthcare setting [59]. However, there are different applicable models and frameworks in healthcare behaviour that received attention in compliance

research, such as health belief model [88, 89], Locus of Control [90], the theory of reasoned action [91] and medical models [60]. Health belief model (HBM) designed by Rosenstock [89] and pointed out that in order to let patient to either to cure or to prevent a given disease, this patient would have to perceive no major obstacles and difficulties that hindering taking the action. The health belief model has been implemented to improve the efficiency of interventions to alternate patient's behaviour by targeting numerous characteristics of the model's vital constructs [92]. Interventions aim to increase perceived susceptibility and perceived seriousness of a healthcare condition by providing knowledge regarding spreading of disease as well as individualized evaluation of risk and consequences of sickness (such as social, medical and financial consequences). The HBM postulates that people will take action to avoid disease if they consider themselves as vulnerable to a condition (perceived susceptibility), if people believe it would have potentially grave consequences (perceived severity), if people believe that available specific course of action would decrease the susceptibility or severity or lead to other positive results (perceived benefits), and if people perceive few negative attributes associated with health actions (perceived barriers) [93]. According to HBM, higher levels of self-efficacy, perceived severity, and a lower level of perceived barriers were associated with better adherence. Self-efficacy was one of the most important mediating variables affecting antihypertensive adherence [94].

## 3. Proposed hypotheses and theoretical framework

Based on the previous literature review, different hypotheses have been proposed:

**H1**: There is a relationship between service quality and patient satisfaction.

**H1a**: There is a relationship between tangibility and patient satisfaction.

**H1b**: There is a relationship between empathy and patient satisfaction.

**H1c**: There is a relationship between assurance and patient satisfaction.

**H1d**: There is a relationship between reliability and patient satisfaction.

**H1e**: There is a relationship between responsiveness and patient satisfaction.

**H1f**: There is a relationship between financial aspect and patient satisfaction.

**H2:** There is a relationship between service quality and patient loyalty.

**H2a:** There is a relationship between tangibility and patient loyalty.

**H2b:** There is a relationship between empathy and patient loyalty.

**H2c:** There is a relationship between assurance and patient loyalty.

**H2d:** There is a relationship between reliability and patient loyalty.

**H2e:** There is a relationship between responsiveness and patient loyalty.

**H2f:** There is a relationship between financial aspect and patient loyalty.

**H3:** There is a relationship between service quality and medication compliance.

**H3a:** There is a relationship between tangibility and medication compliance.

**H3b:** There is a relationship between empathy and medication compliance.

**H3c:** There is a relationship between assurance and medication compliance.

**H3d:** There is a relationship between reliability and medication compliance.

**H3e:** There is a relationship between responsiveness and medication compliance.

**H3f:** There is a relationship between financial aspect and medication compliance.

**H4:** There is a relationship between patient satisfaction and patient loyalty.

**H5:** There is a relationship between patient satisfaction and medication compliance.

**H6**: patient satisfaction mediates the relationship between service quality and patient loyalty.

**H6a**: patient satisfaction mediates the relationship between tangibility and patient loyalty.

**H6b**: patient satisfaction mediates the relationship between empathy and patient loyalty.

**H6c**: patient satisfaction mediates the relationship between assurance and patient loyalty.

**H6d**: patient satisfaction mediates the relationship between reliability and patient loyalty.

**H6e**: patient satisfaction mediates the relationship between responsiveness and patient loyalty.

**H6f**: patient satisfaction mediates the relationship between financial aspect and patient loyalty.

**H7**: patient satisfaction mediates the relationship between service quality and medication compliance.

**H7a**: patient satisfaction mediates the relationship between tangibility and medication compliance.

**H7b**: patient satisfaction mediates the relationship between empathy and medication compliance.

**H7c**: patient satisfaction mediates the relationship between assurance and medication compliance.

**H7d**: patient satisfaction mediates the relationship between reliability and medication compliance.

**H7e**: patient satisfaction mediates the relationship between responsiveness and medication compliance.

**H7f**: patient satisfaction mediates the relationship between financial aspect and medication compliance.

Fig 1 shows the conceptual framework as well as the hypotheses H1, H2, H3, H4 and H5 that test the direct relationship between service quality, patient satisfaction, patient loyalty and medication compliance. Furthermore, the hypotheses H6 and H7 examine the mediation influence of patient satisfaction on the relationship between service quality and dependent variables (patient loyalty and medication adherence).

## 4. Research methodology

### 4.1 Sampling method and sample size

Patients who come to hospitals at least two times were selected, because one visit is not sufficient for patient to create reliable feedback about the hospital services. Respondents were chosen based on random sampling technique from six private hospitals in Damascus. These hospitals are general services hospitals that provide different diagnostic and the therapeutic medical services such as surgical operations, obstetrics and gynaecology services, paediatric

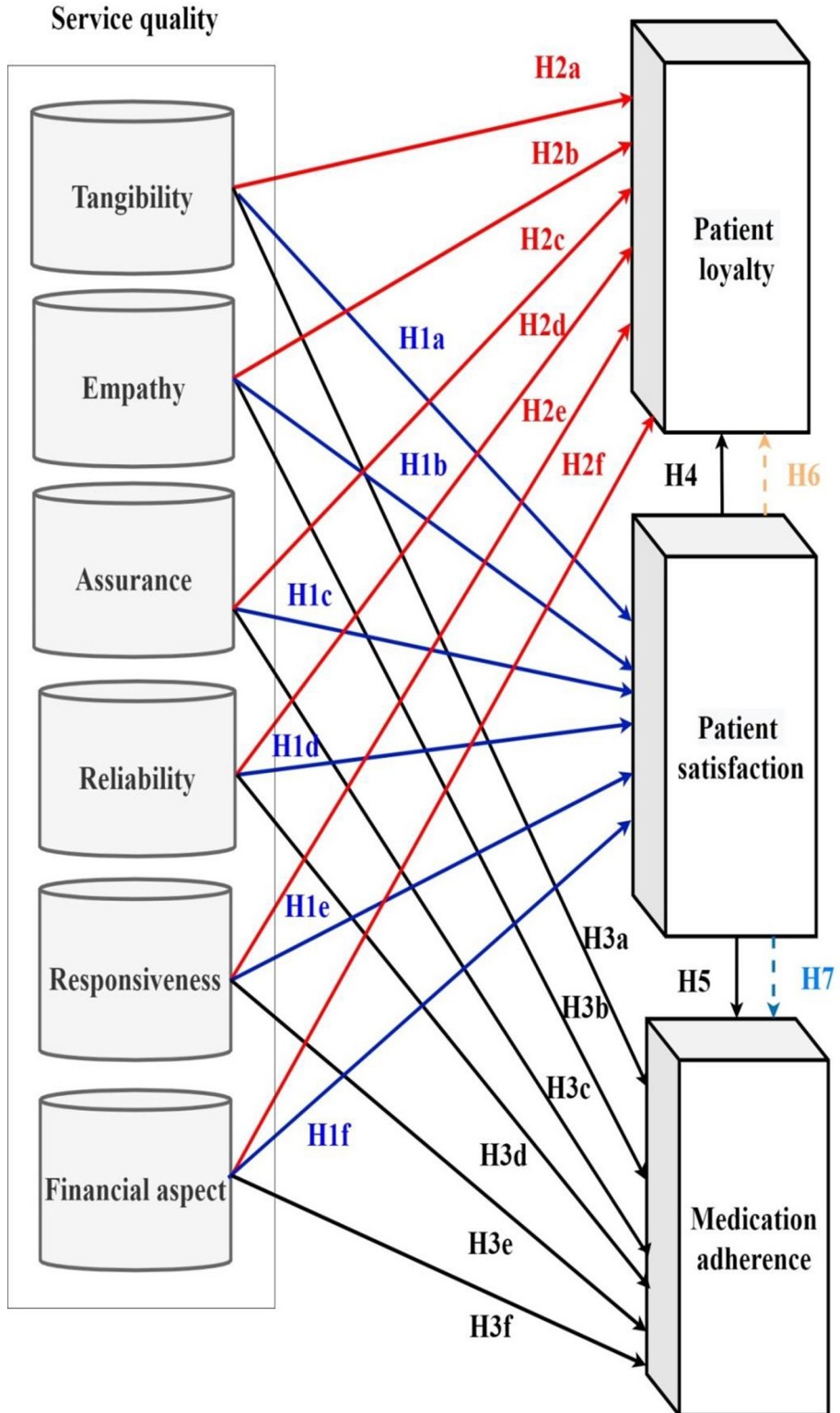

**Fig 1. Conceptual framework that clarifies the model constructs as well as the hypotheses H1, H2, H3, H4, H5, H6, H7.**

services, general medical care, dental services, physiotherapy, occupational therapy, bacteriology analysis services, biochemistry analysis services, medical imaging services, otolaryngology, electrocardiogram diagnostic services, clinical neurophysiology services for electroencephalogram.

During data collection process, we faced many complexities such as reluctance of patients to cooperate. Furthermore, during the COVID-19 pandemic, anxiety, stress levels, depression have increased among patients [95]. Previous studies have also disclosed that patients are hesitated to reveal their feelings and opinions on services provided by their healthcare providers [96]. Firstly, we performed data screening to ensure the accuracy of collected data that includes treatment of missing data and dealing with outliers. We prefer deleting all observations with missing values that leads to reduce variation in the data and causes biases [97]. Besides, we identified very few outliers, and then these outliers were simply removed from the data set. However, we noticed that the deletion had little impact on the structural equation modeling (SEM) findings. Therefore, all possible outliers were kept in this study for further analysis [98]. A total of 600 questionnaires were distributed, the returned were 410 questionnaires with response rate 68.33%. However, 89 questionnaires were excluded because of missing data or not correctly filled [97]. After reviewing the data, 321 questionnaires will be analysed. We had tried to have an equal number of questionnaires from every hospital to avoid any bias.

Several procedural techniques were used to minimise non-response bias in this study, given the data obtained from several hospitals. For example, we informed the participants that their identity and confidentiality were maintained. Also, we informed the patients that the questionnaire is completely anonymous and the researcher will not be able to identify them from the responses provided. Besides, we informed the patients that we would not request sensitive information such as the patient's name anywhere on the questionnaire, and the information collected will only be used for research purposes. In addition, we avoided any long or too difficult questions that would make participants quick to abandon them. Besides, we were sure that there were no psychological issues such as anxiety or depression among the participants. Furthermore, the questionnaire was pretested to eliminate confusing questions and verify that all patients understood every question. A comprehensive set of instructions for completing the survey, including descriptions of the study's construction, were supplied to minimise any confusion. The participants were told there were no right or wrong answers, and the researchers only wanted honest replies.

Table 1 reflects the demographic information of the patients who participated in this study. As depicted in Table 1, there were 106 (33.02%) female and 215 (66.97%) male patients. The majority of the patients came from the age group of 41–50 years old (40.81%). Most of them were single 197 (61.37%) whereas 124 (38.63%) of them were married. In addition, most of the patients were visited the hospital for the second time 211 (65.73%) followed by those who had visited hospital for three times 93 (28.97%).

## 4.2 Questionnaire and measurement instruments

In this research study, based on extensive literature review [16], we proposed a service quality instrument that was originally adapted from SERVQUAL model [12]. The proposed SERVQUAL scale has five dimensions: 1-Tangibles six items (T1, T2, T3, T4, T5, T6), 2-Empathy seven items (E1, E2, E3, E4, E5, E6, E7), 3-Assurance four items (ASSU1, ASSU2, ASSU3, ASSU4, ASSU5), 4-Reliability seven items (REL1, REL2, REL3, REL4, REL5, REL6, REL7), 5-Responsiveness (RES1, RES2, RES3, RES4). However, the SERVQUAL scale was slightly modified with minor alteration in the wording of items to adapt them to healthcare setting and to fit local perceptions (Syrian healthcare setting). Previous studies indicated there is a gap in assessing patient's perception regarding care cost [28–30]. Therefore, there is obviously an

**Table 1. Demographic data of the participated patients.** The data classified based on gender, marital status, age, occupation and number of visiting.

| Gender | Frequency | Percentile |
|---|---|---|
| Male | 215 | 66.97 |
| Female | 106 | 33.03 |
| Marital status | | |
| Single | 197 | 61.37 |
| Married | 124 | 38.63 |
| Age | | |
| 20–30 | 60 | 18.69 |
| 31–40 | 80 | 24.92 |
| 41–50 | 131 | 40.81 |
| 51–60 | 35 | 10.90 |
| above 60 | 15 | 4.67 |
| Occupation | | |
| Student | 23 | 7.17 |
| Government Employee | 60 | 18.69 |
| Private Employee | 91 | 28.35 |
| Self Employed | 120 | 37.38 |
| others | 27 | 8.41 |
| Number of visiting | | |
| Two times | 211 | 65.73 |
| Three times | 93 | 28.97 |
| More than three times | 17 | 5.30 |

essential need for further research to uncover the influence of financial aspect on healthcare outcome. Financial aspect dimension was added to the proposed SERVQUAL model, which has five items (FA1, FA2, FA3, FA4, FA5), and adapted from Marshall et al. and Sumaedi et al. [50, 99]. Patient satisfaction construct has two items (PS1, PS2), which adapted from Wu et al. [100]. Patient loyalty construct has two items (PL1, PL2), which adapted from Kim et al. [101]. Medication adherence construct has four items (MA1, MA2, MA3, MA4), which basically adapted from Lin and Hsieh [102]. As a result, our designed questionnaires that consists of (42) items were administrated to patients in the selected six hospitals to evaluate healthcare service quality, patient satisfaction, patient loyalty and medication compliance. For reducing the language bias, the questionnaire was translated into Arabic language. The English version of the questionnaire was firstly translated to the Arabic language by certified translator. The Arabic version of the questionnaire translated back into English by another translator. Their result was compared with the original questionnaire. Then, the final translated Arabic version was given to two academics, two doctors, three hospital administrator and one nurse. After the translated version accepted by all of them, the questionnaire was ready to distribute. All items were measured on a five-point Likert scale from strongly disagree (1) to strongly agree (5) [103, 104]. To avoid any bias in this quantitative research all the selected hospitals provide the same diagnostic and the therapeutic medical services. Besides, we chose the top-performing hospitals that have the same capacity and services, located in the Syrian capital Damascus.

## 5. Model assessment

The SmartPLS version 3.2.7 [105] was utilized to evaluate structural model and measurement model.

## 5.1 Assessment of measurement model

There are two steps to assess the measurement model; the first step is by testing reliability and the second is to test the validity (convergent validity and discriminant validity). Construct validity affirms to how well the results obtained from the use of the measure fit the concepts around which the investigation is designed. This will be done by conducting the convergent and discriminant validity tests. Convergent validity is the degree to which several items that have been used in measuring the same concept. Outer loadings and average variance extracted (AVE) will be used to measure convergent validity. Reliability test will be utilized to assess the degree of internal consistency in measuring the instruments (questionnaires). It will determine the stability of the instrument in yielding same results each time they are used. Reliability can be tested using the calculation of Cronbach alpha for every variable. The assessment of measurement model includes examination of internal consistency reliability (CR), convergent validity (CV), and discriminant validity (DV). As a result, three items from empathy dimension, one item from reliability dimension, and one item from responsiveness dimension were dropped due to not meeting the criteria: (outer loading >0.7), (AVE > 0.5) or (Cronbach alpha >0.7). Composite reliability (CR) is a better measure of internal consistency compared to Cronbach alpha [106]. All composite reliability (CR) scores exceed the critical value (0.6). Besides, Collinearity issue is evaluated by examining the variance inflated factor (VIF) values of all constructs in the proposed model. The variance inflation factor (VIF) values of all items meet the criteria (not be greater than 5). Hence, the multicollinearity is not a critical issue for the proposed model [107]. Table 2 shows the results of multicollinearity analysis and the outer loading of service quality, patient satisfaction, patient loyalty and medication adherence. Table 3 shows the Cronbach's alpha, composite reliability (CR), average variance extracted (AVE) of all constructs. Discriminant validity refers to how unique are the measurement of different constructs. The Fornell-Larcker analysis was implemented to examine the discriminant validity of the proposed model. The result of discriminant analysis illustrated in Table 4.

## 5.2 Assessment of structural model

The SmartPLS was utilized to analyse the structural model. The structural model presents the relationship among the proposed hypothetical constructs. Bootstrapping was employed to test the level and significance of the path coefficients. The path coefficient values of the structural model were estimated using SmartPLS. The ($R^2$) values should be high enough for the model to achieve a minimum level of explanatory power. The structural model illustrated that ($R^2 = 0.544$) for patient loyalty, ($R^2 = 0.352$) for patient satisfaction, and ($R^2 = 0.462$) for medication adherence. Path coefficients of direct relationships and indirect effects were calculated. Bootstrapping with a re-sampling of 5000 was employed to test the level and significance of the path coefficients. The result of structural model assessment for direct effect showed in Table 5. Besides, Fig 2 highlights the result of the structural model using SmartPLS. The findings illustrate the path coefficients, R-square, outer loading of the items. The following section will present and discuss the findings of the tested hypotheses (H1, H2, H3, H4 and H5) that examined the direct relationship between the constructs. Besides, we will discuss the result of the hypotheses H6 and H7 that examined the mediating impact of patient satisfaction (mediator) on the relationships between independent variables (service quality dimensions) and dependent variables (patient loyalty and medication adherence).

## 6. Findings and discussions

### 6.1 Relationship between service quality and patient satisfaction

This study examined the direct relationship between service quality dimensions and patient satisfaction. As depicted in Table 5, the findings indicated that tangibility ($\beta = 0.179$, $P<0.05$),

**Table 2. Outer loading and calculations of VIF of all items for service quality, patient satisfaction, patient loyalty and medication adherence.**

| Construct | Items | VIF | Outer loading |
|---|---|---|---|
| Assurance | ASSU1 | 1.74 | 0.764 |
| | ASSU2 | 2.805 | 0.876 |
| | ASSU3 | 2.897 | 0.858 |
| | ASSU4 | 2.438 | 0.833 |
| | ASSU5 | 2.53 | 0.86 |
| Empathy | E1 | 2.01 | 0.85 |
| | E2 | 1.965 | 0.847 |
| | E3 | 1.602 | 0.751 |
| | E4 | 1.704 | 0.787 |
| Financial Aspect | FA1 | 3.27 | 0.892 |
| | FA2 | 2.975 | 0.875 |
| | FA3 | 2.357 | 0.845 |
| | FA4 | 1.802 | 0.765 |
| | FA5 | 1.514 | 0.73 |
| Medication adherence | MA1 | 2.263 | 0.816 |
| | MA2 | 2.332 | 0.819 |
| | MA3 | 1.914 | 0.83 |
| | MA4 | 1.845 | 0.815 |
| Patient loyalty | PL1 | 1.619 | 0.888 |
| | PL2 | 1.619 | 0.911 |
| Patient satisfaction | PS1 | 1.563 | 0.893 |
| | PS2 | 1.563 | 0.896 |
| Reliability | REL1 | 2.994 | 0.832 |
| | REL2 | 2.968 | 0.835 |
| | REL3 | 2.419 | 0.836 |
| | REL4 | 3.373 | 0.877 |
| | REL5 | 3.014 | 0.854 |
| | REL6 | 2.914 | 0.845 |
| Responsive | RES1 | 2.724 | 0.893 |
| | RES2 | 2.897 | 0.904 |
| | RES3 | 1.599 | 0.826 |
| Tangibility | T1 | 2.769 | 0.847 |
| | T2 | 3.172 | 0.874 |
| | T3 | 3.36 | 0.888 |
| | T4 | 3.51 | 0.896 |
| | T5 | 1.696 | 0.703 |
| | T6 | 2.141 | 0.806 |

assurance ($\beta = 0.201$, $P<0.01$), reliability ($\beta = 0.132$, $P<0.01$) and financial aspect ($\beta = 0.302$, $P<0.001$) had a positive influence on patient satisfaction. While, empathy ($\beta = 0.042$, $P>0.05$) and responsiveness ($\beta = 0.012$, $P>0.05$) had no significant impact on patient satisfaction. Therefore, H1 hypothesis was rejected that service quality (all six dimensions) had a significant influence on patient satisfaction. However, hypotheses H1a, H1c, H1d and H1f were supported, whereas, hypotheses H1b and H1e were rejected. Our results highlighted that hospital administrators and medical professionals should pay more attention on tangibility, assurance, reliability and financial aspect dimensions, in order to satisfy patients in Syrian hospitals.

**Table 3. Result of statistical analysis: Cronbach's alpha, rho-A, composite reliability (CR), average variance extracted (AVE) for independent variable (service quality dimensions), mediator (patient satisfaction) and dependent variables (patient loyalty, medication adherence).**

| Construct | Cronbach's Alpha | rho_A | Composite Reliability | Average Variance Extracted (AVE) |
|---|---|---|---|---|
| Assurance | 0.894 | 0.896 | 0.922 | 0.704 |
| Empathy | 0.825 | 0.834 | 0.884 | 0.656 |
| Financial aspect | 0.88 | 0.883 | 0.913 | 0.679 |
| Medication adherence | 0.839 | 0.842 | 0.892 | 0.673 |
| Patient loyalty | 0.764 | 0.77 | 0.894 | 0.809 |
| Patient satisfaction | 0.75 | 0.75 | 0.889 | 0.8 |
| Reliability | 0.921 | 0.924 | 0.938 | 0.717 |
| Responsive | 0.846 | 0.845 | 0.907 | 0.766 |
| Tangibility | 0.914 | 0.922 | 0.934 | 0.703 |

In the present study, the findings suggested that visually appealing materials associated with the service, neat appearance and clean uniform of hospital employees, up-to-date facilities and modern medical equipment, clean visually attractive and appealing physical facilities, clear and adequate directional signage, proper temperature inside hospital can lead to better patient satisfaction in Syrian hospitals. Our results revealed that, in order to satisfy patients, healthcare providers should make care costs suitable so that patients have to perceive no financial burdens. This result consistent with previous studies that care cost has a significant effect on patient satisfaction [108, 109]. However, our finding contradicted with the results by Rose et al. [110] that care cost (financial dimension) had no significant impact on service quality from patient's perceptions in Malaysians private hospitals. Besides, our findings indicated that the financial aspect had the highest impact ($\beta = 0.302$) on patient satisfaction followed by assurance ($\beta = 0.201$), tangibility ($\beta = 0.179$) and reliability ($\beta = 0.132$).

## 6.2 Relationship between service quality and patient loyalty

This study examined the direct relationship between service quality dimensions and patient loyalty. As illustrated in Table 5, the findings indicated that reliability ($\beta = 0.14$, $P<0.05$) and responsiveness ($\beta = 0.435$, $P<0.01$) had a positive influence on patient loyalty. While, tangibility ($\beta = 0.129$, $P>0.05$), empathy ($\beta = 0.028$, $P>0.05$), assurance ($\beta = 0.062$, $P>0.05$) and financial aspect ($\beta = 0.043$, $P>0.05$) had no significant impact on patient loyalty. Therefore, hypothesis H2 was rejected that service quality (all six dimensions) had a significant direct impact on patient loyalty. However, the hypotheses H2d and H2e were supported, whereas, the hypotheses H2a and H2b, H2c and H2f were rejected. Our results showed that giving

**Table 4. Result of discriminant validity test using the Fornell-Larcker analysis.**

| No. | Construct | 1 | 2 | 3 | 4 | 5 | 6 | 7 | 8 | 9 |
|---|---|---|---|---|---|---|---|---|---|---|
| 1 | Assurance | 0.839 | | | | | | | | |
| 2 | Empathy | 0.503 | 0.81 | | | | | | | |
| 3 | Financial aspect | 0.48 | 0.379 | 0.824 | | | | | | |
| 4 | Medication adherence | 0.394 | 0.469 | 0.514 | 0.82 | | | | | |
| 5 | Patient loyalty | 0.433 | 0.395 | 0.405 | 0.488 | 0.899 | | | | |
| 6 | Patient satisfaction | 0.464 | 0.372 | 0.499 | 0.462 | 0.496 | 0.894 | | | |
| 7 | Reliability | 0.357 | 0.337 | 0.385 | 0.481 | 0.468 | 0.374 | 0.846 | | |
| 8 | Responsive | 0.323 | 0.301 | 0.287 | 0.436 | 0.618 | 0.261 | 0.375 | 0.875 | |
| 9 | Tangibility | 0.492 | 0.839 | 0.35 | 0.453 | 0.444 | 0.4 | 0.358 | 0.339 | 0.838 |

**Table 5. Result of the hypotheses that tested the direct relationships between variables.**

| Construct | Dimension | Relationship | P Values | Path coefficient | Hypothesis | Result |
|---|---|---|---|---|---|---|
| Service quality | Tangibility | Tangibility ➡ Medication adherence | >0.05 | 0.031 | H3a | Rejected |
| | | Tangibility ➡ Patient loyalty | >0.05 | 0.129 | H2a | Rejected |
| | | Tangibility ➡ Patient satisfaction | <0.05 | 0.179 | H1a | Supported |
| | Empathy | Empathy ➡ Medication adherence | <0.05 | 0.187 | H3b | Supported |
| | | Empathy ➡ Patient loyalty | >0.05 | 0.028 | H2b | Rejected |
| | | Empathy ➡ Patient satisfaction | >0.05 | 0.042 | H1b | Rejected |
| | Assurance | Assurance ➡ Medication adherence | >0.05 | 0.035 | H3c | Rejected |
| | | Assurance ➡ Patient loyalty | >0.05 | 0.062 | H2c | Rejected |
| | | Assurance ➡ Patient satisfaction | <0.01 | 0.201 | H1c | Supported |
| | Reliability | Reliability ➡ Medication adherence | <0.01 | 0.196 | H3d | Supported |
| | | Reliability ➡ Patient loyalty | <0.01 | 0.14 | H2d | Supported |
| | | Reliability ➡ Patient satisfaction | <0.01 | 0.132 | H1d | Supported |
| | Responsiveness | Responsiveness ➡ Medication adherence | <0.001 | 0.198 | H3e | Supported |
| | | Responsiveness ➡ Patient loyalty | <0.001 | 0.435 | H2e | Supported |
| | | Responsiveness ➡ Patient satisfaction | >0.05 | 0.012 | H1e | Rejected |
| | Financial Aspect | Financial aspect ➡ Medication adherence | <0.001 | 0.242 | H3f | Supported |
| | | Financial aspect ➡ Patient loyalty | >0.05 | 0.043 | H2f | Rejected |
| | | Financial aspect ➡ Patient satisfaction | <0.001 | 0.302 | H1f | Supported |
| Patient satisfaction | | Patient satisfaction ➡ Medication adherence | <0.01 | 0.151 | H5 | Supported |
| | | Patient satisfaction ➡ Patient loyalty | <0.001 | 0.239 | H4 | Supported |

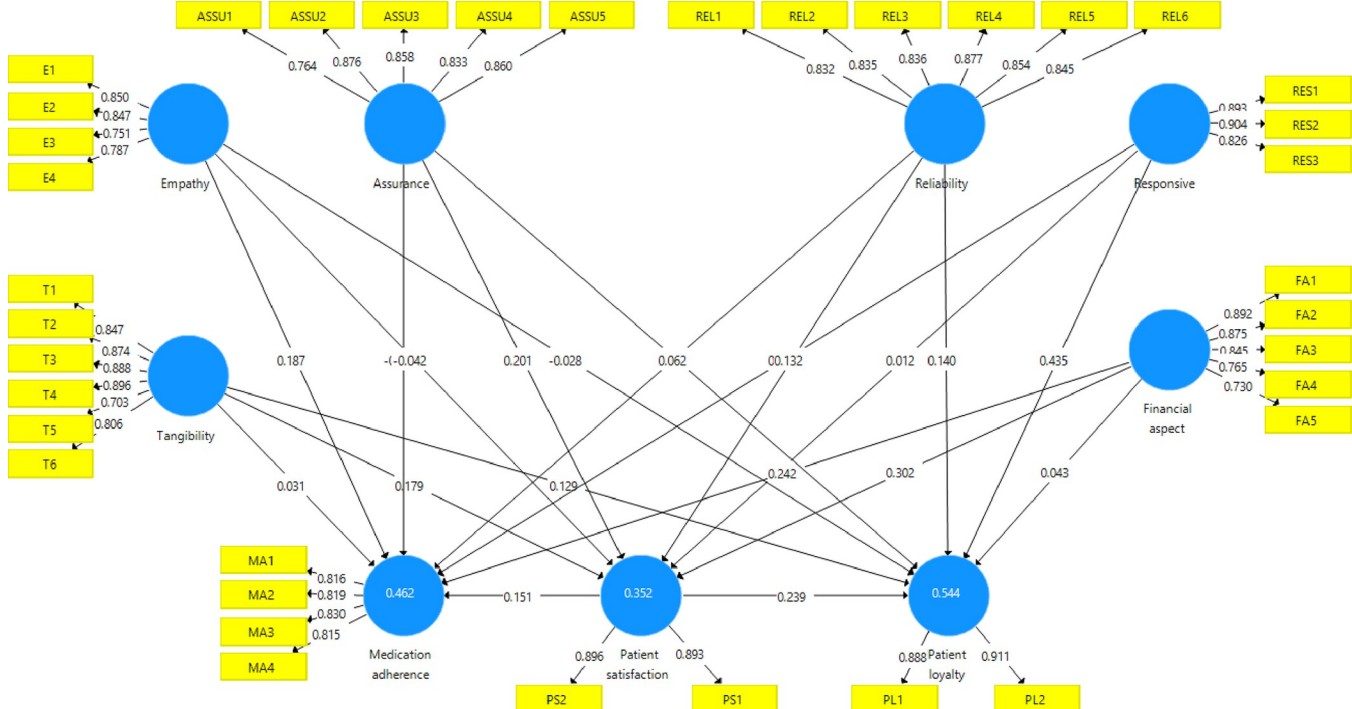

**Fig 2. Result of structural model assessment.** The findings illustrate the path coefficients, R-square, and outer-loading of the items.

prompt service to patients, instilling confidence and trust in patients, and availability to respond to patients' requests are important factors in increasing patient loyalty. However, the findings by other scholars are still mixed and contradictory regarding the influence of service quality on patient loyalty [42, 54]. Our findings indicated that in order to improve patient loyalty in Syrian hospitals, healthcare providers should focus more on reliability and responsiveness dimensions of service quality. Although improving tangibility and assurance dimensions will lead to better patient's satisfaction, but these service quality dimensions have not influence on Syrian patient's loyalty. Therefore, advanced medical equipment, modern physical facilities, and hospital ambiance have no influence on keeping Syrian patients loyal to their healthcare provider. The responsiveness factor had a higher influence than reliability on patient loyalty. Therefore, healthcare providers are always recommended to take the lead in finding the proper help that hospital patients require.

## 6.3 Relationship between service quality and medication adherence

This study examined the direct relationship between service quality dimensions and medication adherence. Based on Table 5, our findings indicated that empathy ($\beta$ = 0.187, $P<0.01$), reliability ($\beta$ = 0.196, $P<0.01$), responsiveness ($\beta$ = 0.198, $P<0.01$), financial aspect ($\beta$ = 0.242, $P<0.01$) had a significant impact on patient's medication adherence. While, tangibility ($\beta$ = 0.031, $P>0.05$) and assurance ($\beta$ = 0.035, $P>0.05$) had no significant relationship with medication adherence. Therefore, hypothesis H3 was rejected that service quality (all six dimensions) had a significant direct impact on medication adherence. In addition, the hypotheses H3b, H3d, H3e and H3f were supported, whereas, the hypotheses H3a and H3c and were rejected. Our findings indicated that in order to improve medication compliance in Syrian hospitals, healthcare providers should pay more attention on empathy, reliability, responsiveness and financial aspect. Shared decision making (reliability) with patient plays an important role in medication compliance and patient satisfaction [111]. Resident doctors frequently should use indirect instructions, scheduling, persuasion, motivation, and direct commands in order to encourage patients to comply with medical advice [112]. The present result supported the findings of the previous studies [7, 113]. They indicated that in order to improve patients' adherence to medication, listening to patients carefully and a good explanation by medical staff are fundamental elements. Besides, spending a sufficient amount of time with patients clarifying the benefits and side effects of taking drugs, prescribing complex drug regimens lead to improve medication adherence [73]. Chandra et al. [67] indicated that a patient-centered approach and trust lead to better care management, improved perceived quality of healthcare services, higher levels of patient satisfaction, and better adherence to treatment. This study is in line with the Social Cognitive Theory (SCT) proposed by Bandura [114] that doctors have to look for a chance to intervention, making a valuable opportunity to help patients realize the type of disease they have and encourage them, in a friendly way based on trust, to take quick action to comply with medication procedures in order to recover quickly. Further, our finding is consistent with previous research by Kripalani et al. [69] that to improve adherence to medication it is required to decrease medication cost. The findings indicated that financial aspect ($\beta$ = 0.242) is the most important factor that had impact on medication adherence, in comparison to the other factors. Besides, empathy, reliability and responsiveness had almost the same direct effect on medication adherence. These results agree with the findings of other studies conducted by (WHO) that healthcare professionals, especially doctors and nurses, need to support, encourage patients, and not blame them during their clinical treatment, in order to help them overcome barriers to medication compliance [115]. Furthermore, our results indicated that doctors in Syrian hospitals should prescribe simple and convenient medications to help

patients to comply with medical recommendations. Besides, our findings are in line with the health belief model (HBM) [89] that, in order to increase medication adherence, doctors should explain the benefits and risks of treatment, give more information about the nature of the illness, and boost patient health literacy.

## 6.4 Relationship between patient satisfaction and patient loyalty

Customer satisfaction has been regarded as an essential element in maintaining and developing long-term and healthy relationship between customers and healthcare professionals [42, 82, 116]. This study examined the direct relationship between patient satisfaction and patient loyalty. Based on Table 5, the findings indicated that patient satisfaction had a significant positive direct influence on patient loyalty (β = 0.239, P<0.001). Therefore, our research finding supports the hypothesis H4 that there is a significant relationship between patient satisfaction and patient loyalty. Customer satisfaction leads to customer retention and willingness to recommend, according to our findings, which are consistent with earlier studies [39, 117]. Similarly, Taneja [118] and Ramli [119] reported that patient satisfaction had a significant impact on patient loyalty. Patient satisfaction is an important concept in determining healthcare services and is considered a major achievement indicator in healthcare organizations [32]. Our present results support the findings that in order to increase patient loyalty in Syrian hospitals, patients need to be satisfied. In addition, our result is in line with expectation confirmation theory (ECT) provided by Oliver [82] that buyers' disconfirmation of service quality is positively associated with their satisfaction, and their satisfaction is positively linked to loyalty intentions [120].

## 6.5 Relationship between patient satisfaction and medication adherence

This study examined the direct relationship between patient satisfaction and medication adherence. As illustrated in Table 5, the findings indicated that patient satisfaction had a significant positive direct influence on medication adherence (β = 0.151, P<0.001). Therefore, H5 was supported by statistical evidence that patient satisfaction significantly had an impact on medication adherence. Our result consistent with previous study by Mohamed and Azizan [65] that there is a significant relationship between behavioural compliance and patient satisfaction. Likewise, previous studies indicated that when patients become dissatisfied with the quality of interaction with medical staff, they are less likely to cooperate with them or even listen to them [71, 72]. Therefore, one of the ways to improve medication adherence is to satisfy patient. Doctors and nurses in Syrian hospitals should try their best to satisfy patients in order to increase their adherence to medical treatment.

## 6.6 Mediating role of patient satisfaction on the relationship between service quality and patient loyalty

Bootstrapping approach was applied to examine possible mediation effect of patient satisfaction on the relationship between service quality and patient loyalty as well as service quality and medication adherence. The results indicated that the bias-corrected 95% percentile confidence interval does not include zero. In Table 6, the findings indicated that patient satisfaction mediated the relationship between service quality dimensions (assurance, reliability and financial aspect) and patient loyalty. Therefore, hypothesis H6c was supported, the indirect effect value is (β = 0.048, P<0.05). Hypothesis H6d was supported, the indirect effect value is (β = 0.032, P<0.05). Hypothesis H6f was supported that patient satisfaction mediated the relationship between financial aspect dimension and patient loyalty. The indirect effect value is (β = 0.072, P<0.05). However, patient satisfaction did not mediate the relationship between patient

**Table 6. Result of the hypotheses that tested the mediating impact of patient satisfaction on the relationship between service quality, patient loyalty and medication adherence.**

| Dependent variable | Hypothesis | | Mediating impact of patient satisfaction | P Values | Specific Indirect Effects | Result |
|---|---|---|---|---|---|---|
| Medication adherence | H7 | H7c | Assurance ➡ Patient satisfaction ➡ Medication adherence | >0.05 | 0.03 | Rejected |
| | | H7b | Empathy ➡ Patient satisfaction ➡ Medication adherence | >0.05 | -0.006 | Rejected |
| | | H7f | Financial aspect ➡ Patient satisfaction ➡ Medication adherence | <0.05 | 0.045 | Supported |
| | | H7d | Reliability ➡ Patient satisfaction ➡ Medication adherence | <0.05 | 0.02 | Supported |
| | | H7e | Responsive ➡ Patient satisfaction ➡ Medication adherence | >0.05 | 0.002 | Rejected |
| | | H7a | Tangibility ➡ Patient satisfaction ➡ Medication adherence | >0.05 | 0.027 | Rejected |
| Patient loyalty | H6 | H6c | Assurance ➡ Patient satisfaction ➡ Patient loyalty | <0.05 | 0.048 | Supported |
| | | H6b | Empathy ➡ Patient satisfaction ➡ Patient loyalty | >0.05 | -0.01 | Rejected |
| | | H6f | Financial aspect ➡ Patient satisfaction ➡ Patient loyalty | <0.01 | 0.072 | Supported |
| | | H6d | Reliability ➡ Patient satisfaction ➡ Patient loyalty | <0.05 | 0.032 | Supported |
| | | H6e | Responsive ➡ Patient satisfaction ➡ Patient loyalty | >0.05 | 0.003 | Rejected |
| | | H6a | Tangibility ➡ Patient satisfaction ➡ Patient loyalty | >0.05 | 0.043 | Rejected |

loyalty and service quality dimensions (tangibility, empathy and responsiveness). Therefore, hypotheses H6a, H6b, H6e were rejected. Our present result supported the findings that healthcare service provider can increase patient loyalty by improving care service (assurance, reliability and financial aspect) provided to satisfied patient. Our finding highlighted that the direct effect of service quality (reliability and financial aspect) has higher impact on patient loyalty in comparison to indirect effect through patient satisfaction. Although, there was no significant direct relationship between assurance and patient loyalty ($\beta = 0.129$, $P>0.05$), but patients satisfaction mediated the relationship between assurance and loyalty ($\beta = 0.086$, $P<0.05$). Therefore, healthcare providers can increase patient loyalty through improving assurance dimension of care services to satisfied patient. This study reinforces the finding by Guo et al [121] that improving the quality of medical services is the main path to acquire patient loyalty for private hospitals.

## 6.7 Mediating role of patient satisfaction on the relationship between service quality and medication adherence

In Table 6, the findings indicated that patient satisfaction mediated the relationship between service quality dimensions (reliability and financial aspect) and medication adherence. Therefore, hypothesis H7d was supported (the indirect effect value is 0.02). Furthermore, hypothesis H7f was supported (the indirect effect value is 0.045). However, patient satisfaction did not mediate the relationship between medication adherence and service quality dimensions (tangibility, empathy, assurance and responsiveness). Therefore, hypotheses H6a, H6b, H6c and H6e were rejected. Although, there was no mediating effect on the relationship between medication adherence and service quality dimensions (assurance and empathy), but assurance and empathy had a direct relationship on medication adherence. Our finding contradicted with the findings by Mohamed and Azizan [122] that the indirect effect of patient satisfaction has better influence on the relationship between service quality and compliance than the direct effect. Our finding highlighted that the direct effect of service quality (reliability and financial aspect) has higher impact on medication compliance in comparison to indirect effect through patient satisfaction. Our present result supported the findings that healthcare service provider can increase medication adherence by improving care service (reliability and financial aspect) provided to satisfied patient. Table 5 summarized the result of the hypotheses that tested the direct relationship between service quality dimensions (tangibility, empathy, assurance, reliability,

responsiveness and financial aspect), patient satisfaction (mediator), patient loyalty (dependant variable), and medication adherence (dependant variable). As a result, the hypotheses H1a, H1c, H1d, H1f, H2d, H2e, H3b, H3d, H3e, H3f, H4 and H5 were supported. Table 6 summarized the results of the tested hypotheses that examined the mediating effect of patient satisfaction on the relationship between service quality dimensions and dependent variables (patient loyalty and medication adherence). As a result, H6c, H6d, H6f, H7d and H7f were supported.

## 7. Research implications and recommendations

This study has gone some way towards improving our understanding of the mediating effect of patient satisfaction on the relationship between service quality, patient loyalty and medication adherence in Syrian healthcare setting. Our conceptual framework improves hospital administrator's knowledge about the essentials determinants (dimensions) of service quality and provides a deep understanding about how service quality influences patient satisfaction, patient loyalty and medication adherence. Our result highlighted that in order to satisfy Syrian patients in private hospitals, doctors, nurses and hospital administrators should pay attention on tangibility, assurance, reliability and financial aspect dimensions. This finding has important implications for improving the patient loyalty in the Syrian hospitals. The healthcare providers in Syria can gain and maintain their patient's loyalty through focusing on reliability and responsiveness dimensions of service quality.

Courteous behaviour is just one method to assist medical professionals seem more approachable and likeable. Moreover, the behaviour of doctors (cooperative, friendly, supportive, courteous) is a vital determinant of patients' satisfaction. Therefore, patients prefer selecting healthcare providers that offer better service quality and have medical professional staff with courteous behaviour [123, 124]. In addition, a warm communication by medical staff especially nurses at reception is an important satisfaction determinant [125].

One of most interesting findings is that financial aspect in Syrian healthcare is an essential element in satisfying patient and increasing the medication adherence to clinical treatment. The financial aspect of healthcare services had the highest impact on Syrian patient satisfaction and on medication adherence. Therefore, policymakers and hospital administrators as well as doctors in Syrian hospitals should do their best to assure that patients perceive no major obstacles and difficulties such as financial burdens that hindering them from medication adherence [89]. The world health organization (WHO) called attention to the high proportions of healthcare spending paid out-of-pocket in most developing countries. Furthermore, the formal and informal out-of-pocket payments have become a major source of health financing. Indeed, increasing the care cost has devastating consequences on lower income groups and undermined the equal access to health care. In Syria, the out-of-pocket expenditure as per cent of total health expenditure and the governments' expenditure on healthcare as per cent of general government expenditure were (53.9%) and (5.3%), respectively [126]. There is a need for the establishment of sustainable healthcare financing systems that ensures universal access to healthcare services.

Praising and rewarding patients for their "attempts" at achieving a behavioural goal are associated with significantly higher self-efficacy and physical activity. Hence, medical professionals, especially doctors, can focus on small successes and progression toward behavioural goals rather than actual achievement of final target behaviours. Besides, healthcare professionals should always encourage patients to comply with their medication treatment. Patient education on perceived benefits and obstacles will provide patients with incentives to involve in health-promoting behaviours. Interventions will assist in lifting up self-efficacy by providing

training in particular health promoting behaviours, mostly for multifaceted lifestyle alterations (for example, altering diet or doing physical activities, comply with a complex medication regimen, physical environment). Educational programs designed by doctors play a crucial role in changing patient's negative behaviour, reducing perceived barriers, that lead to improvement in patient's situation [127].

## 8. Limitation and future research direction

Our study gives suggestions and recommendations to improve the healthcare sector in Syria. However, this study is based on patient's perception of provided healthcare services. We ignored the medical professionals' perception (doctors and nurses) that should also be investigated in future.

## 9. Conclusion

The purpose of this research was to empirically examine the relationships between service quality, patient satisfaction, patient loyalty, and medication adherence in the Syrian healthcare setting from a patient's perspective. A quantitative approach has been adopted to gather data from patients in six hospitals located in the Syrian capital, Damascus. After reviewing the data, (321) questionnaires were selected for statistical analysis. The reliability and validity of the theoretical model had been confirmed.

Our proposed model can significantly explain 35 per cent of patient satisfaction, 55 per cent patient loyalty and 46 per cent medication adherence in a statistically manner.

The findings of this study indicated that only four dimensions of service quality (tangibility, assurance, reliability and financial aspects) had a positive influence on the satisfaction of the Syrian patients. However, empathy and responsiveness had no significant impact on patient satisfaction. Therefore, healthcare providers in Syria should pay more attention on tangibility, assurance, reliability and financial aspect in order to satisfy their patients. Financial aspect had the highest impact on patient satisfaction followed by assurance, tangibility and reliability.

Furthermore, our investigation revealed that reliability and responsiveness had a positive impact on patient loyalty. However, tangibility, empathy, assurance and financial aspect had no influence on patient loyalty. As a result, in order to increase patient loyalty in Syrian hospitals, healthcare professionals should place a greater emphasis on the service quality aspects of reliability and responsiveness. Interestingly, in Syrian hospitals, the responsiveness of medical professionals had the highest effect on patient's loyalty. The responsiveness had three times higher influence than reliability dimension on patient loyalty. Therefore, healthcare providers are always recommended to take the lead in finding the proper help that hospital patients require, giving prompt service to the patient, instil confidence and trustworthy in patients, and maintaining constant availability to respond to the patient's request.

In addition, empathy, reliability, responsiveness and financial aspect, had a significant impact on patient's medication adherence. However, tangibility and assurance dimensions had no significant influence on the medication adherence. Our finding supports the fact that hospital's employees, especially doctors and nurses, should focus on empathy, reliability, responsiveness, care cost in order to increase patient's medication adherence. Listening and explaining skills of medical professionals will help patients to comply with their clinical treatment. Providing services at the time they promise to do, sharing decision with patients and their family, keeping accurate record and billing, prescribing a simple and effective regimen, encouraging patients to comply with medical advice, giving prompt service to the patient are essential factors to improve medication adherence. Financial aspect considered as the most influential factor that affects medication adherence. Hence, policy makers, hospital's

administrators, healthcare professionals should help patients to improve their compliance behaviour by decreasing the medication treatment fees. Empathy, reliability and responsiveness had almost the same direct effect on medication adherence.

Besides, our findings highlighted that patient satisfaction had a significant positive direct influence on patient loyalty. The findings of this study revealed that in order to increase patient loyalty in Syrian hospitals, patient needs to be satisfied. The findings indicated that patient satisfaction had a significant positive direct influence on medication adherence. Therefore, one of the ways to improve medication adherence is to satisfy patient. Doctors and nurses should satisfy Syrian patients in order to increase their adherence to medical treatment.

Our results indicated that patient satisfaction mediated the relationship between service quality dimensions (assurance, reliability and financial aspect) and patient loyalty. However, patient satisfaction did not mediate the relationship between patient loyalty and service quality dimensions (tangibility, empathy and responsiveness). Although, there was no significant direct relationship between assurance and patient loyalty, but patients' satisfaction mediated the relationship between assurance and loyalty. Therefore, healthcare providers can increase patient loyalty through improving assurance dimension of care services to satisfied patient. Our present result supported the findings that healthcare service provider can also increase patient loyalty by improving service quality (reliability and financial aspect) to satisfied patient.

Our study illustrated that patient satisfaction mediated the relationship between service quality dimensions (reliability and financial aspect) and medication adherence. However, patient satisfaction did not mediate the relationship between medication adherence and service quality dimensions (tangibility, empathy, assurance and responsiveness). Our present result supported the fact that healthcare service provider can increase medication adherence by improving care service (reliability and financial aspect) provided to satisfied patient.

Another important finding was that reliability is the only dimension of service quality that has a significant direct impact on patient satisfaction, patient loyalty and medication adherence. However, the lowest score among the service quality dimensions was the reliability dimension. As a result, additional investigation is needed to look at the causes of the poor reliability score from the patient's perspective. Empathy had only a significant impact on medication compliance, while, assurance had only a significant influence on patient satisfaction. Furthermore, tangibility had only significant impact on patient satisfaction, but tangible dimension had no influence on patient loyalty and mediation adherence.

The most interesting finding was that in the Syrian healthcare setting, the financial impact had the highest impact on patient satisfaction and on medication adherence.

## Supporting information

**S1 File.**
(PDF)

## Author Contributions

**Conceptualization:** Firas AlOmari, Abu Bakar A. Hamid.

**Data curation:** Firas AlOmari, Abu Bakar A. Hamid.

**Formal analysis:** Firas AlOmari, Abu Bakar A. Hamid.

**Investigation:** Firas AlOmari, Abu Bakar A. Hamid.

**Methodology:** Firas AlOmari, Abu Bakar A. Hamid.

**Project administration:** Firas AlOmari, Abu Bakar A. Hamid.

**Resources:** Firas AlOmari, Abu Bakar A. Hamid.

**Software:** Firas AlOmari, Abu Bakar A. Hamid.

**Supervision:** Firas AlOmari, Abu Bakar A. Hamid.

**Validation:** Firas AlOmari, Abu Bakar A. Hamid.

**Visualization:** Firas AlOmari, Abu Bakar A. Hamid.

**Writing – original draft:** Firas AlOmari, Abu Bakar A. Hamid.

**Writing – review & editing:** Firas AlOmari, Abu Bakar A. Hamid.

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
