## [Decision Letter · Decision Letter 0]

1 Jun 2022

PONE-D-21-25433Strategies to Improve Patient Loyalty and Medication Adherence in Syrian Healthcare Setting: The Mediating Role of Patient SatisfactionPLOS ONE

Dear Dr. AlOmari,

Thank you for submitting your manuscript to PLOS ONE. After careful consideration, we feel that it has merit but does not fully meet PLOS ONE’s publication criteria as it currently stands. Therefore, we invite you to submit a revised version of the manuscript that addresses the points raised during the review process. Your manuscript has undergone the peer-review process and the reviewers have provided their comments/suggestions. Kindly address these points/concerns before we make a decision. Please submit your revised manuscript by Jul 16 2022 11:59PM. If you will need more time than this to complete your revisions, please reply to this message or contact the journal office at plosone@plos.org. Please include the following items when submitting your revised manuscript:A rebuttal letter that responds to each point raised by the academic editor and reviewer(s). You should upload this letter as a separate file labeled 'Response to Reviewers'.A marked-up copy of your manuscript that highlights changes made to the original version. You should upload this as a separate file labeled 'Revised Manuscript with Track Changes'.An unmarked version of your revised paper without tracked changes. You should upload this as a separate file labeled 'Manuscript'.If applicable, we recommend that you deposit your laboratory protocols in protocols.io to enhance the reproducibility of your results. Protocols.io assigns your protocol its own identifier (DOI) so that it can be cited independently in the future. For instructions see: https://journals.plos.org/plosone/s/submission-guidelines#loc-laboratory-protocols. Additionally, PLOS ONE offers an option for publishing peer-reviewed Lab Protocol articles, which describe protocols hosted on protocols.io. Read more information on sharing protocols at https://plos.org/protocols?utm_medium=editorial-email&utm_source=authorletters&utm_campaign=protocols.

We look forward to receiving your revised manuscript.

Kind regards,

Kingston Rajiah

Academic Editor

PLOS ONE

Journal Requirements:

2. Please provide additional details regarding participant consent. In the Methods section, please ensure that you have specified (1) whether consent was informed and (2) what type you obtained (for instance, written or verbal). If your study included minors, state whether you obtained consent from parents or guardians. If the need for consent was waived by the ethics committee, please include this information.

4. Thank you for stating the following financial disclosure: "The funders had no role in study design, data collection and analysis, decision to publish, or preparation of the manuscript." 

Reviewers' comments:

Reviewer's Responses to Questions

**Comments to the Author**

1. Is the manuscript technically sound, and do the data support the conclusions?

Reviewer #1: Yes

Reviewer #2: Yes

2. Has the statistical analysis been performed appropriately and rigorously? 

Reviewer #1: Yes

Reviewer #2: Yes

3. Have the authors made all data underlying the findings in their manuscript fully available?

Reviewer #1: Yes

Reviewer #2: No

4. Is the manuscript presented in an intelligible fashion and written in standard English?

Reviewer #1: No

Reviewer #2: Yes

5. Review Comments to the Author

Reviewer #1: The research is well-designed and served purpose of the study. The aim of study was to examine the relationships between service quality, patient satisfaction, patient loyalty and medication adherence in the Syrian healthcare system.

The work is well-written however; some minor changes are required before acceptance:

• In literature review, compare your work with reported work, highlighting main advantages of your work.

• Conclusion is not comprehensive; it should be precise and targeted the major results obtained.

• In research methodology, a total of 600 questionnaires were distributed, the returned were (410) questionnaires with response rate (68.33%)…….. Why 410 and 68.33% are mentioned in bracket? It makes sense in this sentence…… As depicted in Table 1, there were 106 (33.02%) female and 215 (66.97%) male patients. In all other places modify it.

• Exclusion criteria should be better explained.

• Research implications and recommendations section is too lengthy. Make it short with relevant data.

Reviewer #2: This article is well researched, well written and well analysed. Most details have been explained well. The only problem with the article is a little bit of grammatical mistakes. Other than that it is very fine work.

6. PLOS authors have the option to publish the peer review history of their article (what does this mean?). If published, this will include your full peer review and any attached files.

Reviewer #1: No

Reviewer #2: **Yes: **Yash Alok

---

## [Author Response · Author response to Decision Letter 0]

9 Jul 2022

Dear Editor-in-Chief,

Thank you for giving the opportunity to revise our manuscript entitled “Strategies to improve patient loyalty and medication adherence in Syrian healthcare setting: The mediating role of patient satisfaction”. The manuscript had been revised based on the suggestions given by the Reviewers. 

We appreciate the interest that Editor has taken in the submitted manuscript and the constructive comments He has given. However, we will do our utmost to answer any other questions from the Reviewers. 

The authors express profound gratitude for providing a full publication fee waiver offered by PLOS. 

Regards

Authors

Answers- Reviewer A- Comment:

Author would like to take this opportunity to thank Reviewer for the effort and expertise to review manuscript. Author had rewritten the manuscript with accuracy, brevity and clarity. The manuscript had been organized properly, keeping in mind fully incorporate reviewer’s suggestions into a revised manuscript and respect the requirements of the journal.

Answers- Reviewer A- Comment:

The research is well-designed and served purpose of the study. The aim of study was to examine the relationships between service quality, patient satisfaction, patient loyalty and medication adherence in the Syrian healthcare system.

The work is well-written however; some minor changes are required before acceptance:

• In literature review, compare your work with reported work, highlighting main advantages of your work.

Recent previous literatures were added as a foundation and as support for a new insight that we contribute for (2.1 relationship between SQ, PS and PL) and for (2.2 relationship between SQ, PS and MA). The literature reviews also provide a solid background for our research paper’s investigation.

We also highlighted the significance of our research work by clarifying and explaining the main gaps within a specific subject area. For example, the modification of SERVQUAL model, the reason behind including the financial aspect component (we summarize and synthesize the arguments and ideas of the scholars). The mediating impact of PS on the relationship between SQ and MA. 

• Conclusion is not comprehensive; it should be precise and targeted the major results obtained.

The conclusion section had been improved to include all the main results (based on the findings of the proposed hypotheses). 

• In research methodology, a total of 600 questionnaires were distributed, the returned were (410) questionnaires with response rate (68.33%)…….. Why 410 and 68.33% are mentioned in bracket? It makes sense in this sentence…… As depicted in Table 1, there were 106 (33.02%) female and 215 (66.97%) male patients. In all other places modify it.

As requested, the brackets were removed from this part: A total of 600 questionnaires were distributed, the returned were 410 questionnaires with response rate 68.33%. However, 89 questionnaires were excluded because of missing data or not correctly filled. While, the brackets in this paragraph were kept (the number outside the bracket reflects the frequency, while the number inside the brackets reflects the percentile): Table 1 reflects the demographic information of the patients who participated in this study. As depicted in Table 1, there were 106 (33.02%) female and 215 (66.97%) male patients. The majority of the patients came from the age group of 41-50 years old (40.81%). Most of them were single 197 (61.37%) whereas 124 (38.63%) of them were married. In addition, most of the patients were visited the hospital for the second time 211 (65.73%) followed by those who had visited hospital for three times 93 (28.97%).

• Exclusion criteria should be better explained.

The exclusion criteria were clarified: During data collection….; Besides, we identified very few outliers, then….; Several procedural techniques were used to minimize non-response….; Besides, we informed the patients…. 

• Research implications and recommendations section is too lengthy. Make it short with relevant data.

Done. 

Answers- Reviewer B- Comment:

Author would like to take this opportunity to thank Reviewer for the effort and expertise to review manuscript. Author had rewritten the manuscript with accuracy, brevity and clarity. The manuscript had been organized properly, keeping in mind fully incorporate reviewer’s suggestions into a revised manuscript and respect the requirements of the journal.

---

## [Editor Report · Decision Letter 1]

13 Jul 2022

Strategies to Improve Patient Loyalty and Medication Adherence in Syrian Healthcare Setting: The Mediating Role of Patient Satisfaction

PONE-D-21-25433R1

Dear Dr. AlOmari,

We’re pleased to inform you that your manuscript has been judged scientifically suitable for publication and will be formally accepted for publication once it meets all outstanding technical requirements.

Kind regards,

Kingston Rajiah

Academic Editor

PLOS ONE
---

## [Editor Report · Acceptance letter]

10 Nov 2022

PONE-D-21-25433R1 

Strategies to improve patient loyalty and medication adherence in Syrian healthcare setting: The mediating role of patient satisfaction 

Dear Dr. AlOmari:

I'm pleased to inform you that your manuscript has been deemed suitable for publication in PLOS ONE. Congratulations! Your manuscript is now with our production department. 

Kind regards, 

on behalf of

Associate Professor Kingston Rajiah 

Academic Editor

PLOS ONE